# Bromodomain and Extra-Terminal Proteins in Brain Physiology and Pathology: BET-ing on Epigenetic Regulation

**DOI:** 10.3390/biomedicines11030750

**Published:** 2023-03-01

**Authors:** Noemi Martella, Daniele Pensabene, Michela Varone, Mayra Colardo, Michele Petraroia, William Sergio, Piergiorgio La Rosa, Sandra Moreno, Marco Segatto

**Affiliations:** 1Department of Biosciences and Territory, University of Molise, Contrada Fonte Lappone, 86090 Pesche, Italy; 2Department of Science, University Roma Tre, Viale Marconi 446, 00146 Rome, Italy; 3Laboratory of Neurodevelopment, Neurogenetics and Neuromolecular Biology, IRCCS Santa Lucia Foundation, 64 via del Fosso di Fiorano, 00179 Rome, Italy; 4Division of Neuroscience, Department of Psychology, Sapienza University of Rome, via dei Marsi 78, 00185 Rome, Italy

**Keywords:** CNS, BRD2, BRD3, BRD4, neurodevelopment, memory, behavior, neurodegeneration, neuroinflammation, neuropsychiatric disorders

## Abstract

BET proteins function as histone code readers of acetylated lysins that determine the positive regulation in transcription of genes involved in cell cycle progression, differentiation, inflammation, and many other pathways. In recent years, thanks to the development of BET inhibitors, interest in this protein family has risen for its relevance in brain development and function. For example, experimental evidence has shown that BET modulation affects neuronal activity and the expression of genes involved in learning and memory. In addition, BET inhibition strongly suppresses molecular pathways related to neuroinflammation. These observations suggest that BET modulation may play a critical role in the onset and during the development of diverse neurodegenerative and neuropsychiatric disorders, such as Alzheimer’s disease, fragile X syndrome, and Rett syndrome. In this review article, we summarize the most recent evidence regarding the involvement of BET proteins in brain physiology and pathology, as well as their pharmacological potential as targets for therapeutic purposes.

## 1. Introduction

In 1992, the identification of bromodomain (BrD) as a highly conserved protein motif in human, *Drosophila,* and yeast proteins constituted a considerable step towards the advancement of knowledge on the molecular mechanisms that modulate gene expression [1]. To date, 46 BrD-containing proteins, encoded by the human genome, have been identified [2,3]. BrD is present in epigenetic readers capable of recognizing histone acetylation on lysine (Kac) residues. Among the post-translational modifications of histones, Kac extensively affects chromatin structure and function as it slackens nucleosome packaging within chromatin, increasing DNA accessibility [4]. The interaction of BrDs with Kac determines the recruitment of epigenetic protein complexes which alter gene expression by modulating the accessibility of chromatin to transcription factors (TFs) [5,6,7]. Therefore, it is not surprising that alterations in several BrD-containing proteins have been functionally described in disease processes, including cancer, inflammation, and viral replication [8,9,10]. Among the different BrD-containing epigenetic modulators, the family of proteins containing the bromodomain and the extra-terminal domain (BET) has aroused interest in biomedical research, representing new and interesting therapeutic targets. In the last decade, the understanding of the molecular pathways involving BET proteins has been greatly accelerated by the discovery of selective inhibitors interfering with the interaction between their BrDs and the acetylated histones [11,12]. The involvement of this class of epigenetic readers has been primarily characterized in different types of cancer and inflammatory contexts, due to their role in the transcriptional modulation of oncogenes and immune response mediators [13,14]. However, recent studies highlighted that BET proteins may also play key roles in central nervous system (CNS) homeostasis, regulating the expression of several genes including neurotransmitter receptors, ion channels, and neurotrophic factors [15,16,17,18,19]. Consistently, BET proteins exhibit a high affinity for histone modifications associated with learning and memory processes [20,21,22]. Furthermore, pharmacological inhibition of BET proteins regulates the expression of multiple genes associated with neuroplasticity and cognition [15,16,18,23]. In this article, we critically review the up-to-date body of knowledge concerning the epigenetic modulation by BET proteins in the physiological processes regulating CNS homeostasis. We also address the relationship linking BET proteins’ altered activity and brain disorders and their pharmacological modulation as a prospective tool, opening novel therapeutic avenues.

## 2. Structure and Functions of BET Proteins

Histone acetylation is the most relevant post-translational modification [24], as it induces the recruitment of TFs and chromatin remodeling factors that favor the transcriptional process. These factors are recruited by bromodomain-containing proteins (BRDs), where the BrD acts as an epigenetic reader domain that specifically recognizes Kac residues [25]. The human proteome has 46 BRDs, for a total of 61 BrDs identified. Each BrD is a motif of about 110 amino acids and the first atomic structure was defined by the laboratory of Ming-Ming Zhou [26]. The characteristic and conserved structure of BrD comprises four α-left-handed helices, named αZ, αA, αB, and αC, connected by two different loops defined as ZA loop and BC loop, respectively [3]. Within the domain, two highly conserved residues have been identified among the BRDs. These are a Tyr residue in the ZA loop which stabilizes folding and an Asn residue located in the BC loop which favors docking of Kac [25]. These residues establish water-mediated and hydrogen bonds with the acetyl group of Kac. Therefore, the structural topology of BrD is defined in a small hydrophobic pocket made up of the four α-helices and the ZA and BC loops, which host the side chain of Kac. Human BRDs have been classified into nine large families based on sequence, structural, and functional homologies [3,5]. BET proteins belong to the fifth group of the BRD family. In recent years, BET proteins have attracted increasing interest since, as transcriptional coactivators, they clearly contribute to the onset and progression of numerous pathological conditions like cancer, diabetes, inflammation diseases, renal diseases, neurodegenerative disorders, and many other diseases [27,28]. Furthermore, attention towards BETs has increased with the discovery of small inhibiting molecules, termed as BET inhibitors (BETis).

The BET protein group consists of four members: BrD-containing protein 2 (BRD2), BRD3, and BRD4, which are ubiquitous in the human organism, and the testis-specific bromodomain-containing protein (BRDT) [29,30]. Structurally, BETs have two BrDs arranged in tandem, named BrD1 and BrD2, and an extra-terminal domain (ETD). BrD1 consists of the following amino acid residues: Gln85, ASP144, Lys141, and Ile146. On the other hand, BrD2 is composed of Lys374, Val435, Pro430, and His433 [12,31,32]. The BrDs of BET proteins have moderate affinity for mono-acetylated lysine residues, while their residual affinity is high for regions of 1–5 amino acids with multiple acetylation sites [33,34]. BET proteins promote the transcription by facilitating the opening of chromatin and by recruiting coactivators and TFs on gene promoters and enhancers. Furthermore, they promote transcriptional elongation by activating the RNA polymerase II (RNApol-II) complex [35]. The best-studied BET member is BRD4. Its role as transcriptional regulator is mediated through association with protein complexes such as positive transcriptional elongation factor b (P-TEFb) [36,37,38], a heterodimer consisting of the cyclin-dependent kinase Cdk9 and regulatory subunits such as Cyclin T1, T2, or K [39]. Two BRD4 regions are directly involved in P-TEFb binding: the C-terminal domain (CTD), which interacts with Cyclin T1 and with Cdk9, and BrD2, which recognizes an acetylated region of Cyclin T1 [37,40,41]. Before being recruited by BRD4, P-TEFb is retained in an inactive state by the 7SK/HEXIM ribonucleoprotein complex [37,42]. Specifically, besides recruiting P-TEFb, BRD4 mediates the phosphorylation of the active motif of Cdk9 [43], which in turn suppresses the inhibitory function of the regulatory factors DSIF and NELF [39,44]. Subsequently BRD4, in association with P-TEFb, displaces the 7SK/HEXIM complex and, acting as an atypical kinase, phosphorylates the Ser2 in the C-terminal region of RNApol-II [39,45,46]. Furthermore, BRD4 modulates the activity of the RNApol-II transcriptional coactivator known as the “mediator complex” [47]. It is worth noting that the functional interaction between BET and RNApol-II is not limited to BRD4. In fact, BRD2 and BRD3 also favor the elongation activity of the enzyme as they can interact with the hyperacetylated chromatin [48]. BRD4 ETD, on the other hand, acts by recruiting transcriptional activators such as histone acetyltransferase (HAT) P300, histone arginine demetylase JMJD6, and histone methyltransferase NSD3. Furthermore, BRD4 is also able to bind other acetylated proteins such as TFs [49]. A kinase activity can also be attributed to BRD4. Indeed, its ETD can bind SWI-SNF and CHD2 which are responsible for ATP-dependent chromatin remodeling [6,50]. Therefore, the binding of BET proteins to Kac at promoter regions favors the recruitment of RNApol-II, activating the transcription of target genes [2] (Figure 1). 

BETs can also modulate transcription through interaction with regulatory regions distant from the promoter. In fact, BRD4 can bind to some particularly large enhancer sequences, known as superenhancer regions, modulating the transcription of downstream genes [51]. A plethora of studies demonstrated the role of BET proteins in the development and progression of different pathologies, ranging from cancer and metabolic disorders to CNS diseases. Indeed, recent studies show that BETs are inhomogeneously expressed in the CNS, being particularly abundant in brain regions involved in reward mechanisms [52]. These proteins play an important role in neurogenesis [53], in neural tube closure defects [54], and in orchestrating the transcriptional regulation underlying learning and memory processes [15,22,55]. Consistent with their pivotal role in the homeostatic maintenance of brain functions, deregulations in their expression/activity are involved in neurodegenerative conditions [23,56] and in neuroinflammation [57,58]. 

The first pan-BET inhibitors discovered were 3-methyltriazolothienodiazepines and methyltriazolobenzodiazepines [11]. Structural studies have shown that members of this class of small molecules bind to the acetyl-lysine binding pocket of BET BrDs. In particular, the triazolo ring functions as an acetyl-lysine mimetic moiety, mimicking the hydrogen bond determined by the acetyl-lysine carbonyl to a conserved asparagine and a water-mediated hydrogen bond to a conserved tyrosine. A specific feature of BET BrDs is the WPF shelf, which is often targeted by aromatic or hydrophobic moieties of BETis, thereby conferring great potency and selectivity to BET domains [59]. JQ1 is the prototype pan-BET inhibitor. Importantly, this compound is still being used successfully for proof-of-concept studies in preclinical research. Following the discovery of JQ1, fragments containing a dimethyl isoxazole core led to the synthesis of new and different classes of pan-BET inhibitors such as iBET151, the sub-nanomolar inhibitor HJB97, and other molecules that have successfully entered clinical trials [60,61]. To date, more than 600 crystal structures of BET BrDs have been registered in the Protein Data Bank (PDB, https://www.rcsb.org/, accessed on 10 January 2023). The different roles of BET proteins in maintaining cellular homeostasis led to speculation that BETis, to be successful in clinical practice, should be selective for a specific BrD. Interestingly, recent studies have shown that selective targeting of BD1 or BD2 is possible and could be an effective strategy to achieve therapeutic potential by limiting off-target effects. For example, RVX-208 has been shown to interact with solvent-exposed residues that are conserved in BD2 but different in BD1. These variations in specific amino acid residues were essential for the development of novel BD2-selective inhibitors such as ABBV-744. On the other hand, the BD1 selectivity of BETis was achieved by optimizing the interaction with the unique aspartate/lysine residues located in the BC loop [60,62]. For more information on BETis, please see the review by Schwalm and Knapp (2022) [60].

Targeting BET proteins by small inhibitors (BETis) is intriguing for many researchers. Among the BETis developed until now, JQ1, OTX015, and I-BET858 are non-covalent and selective inhibitors able to cross the blood–brain barrier (BBB) and produce molecular and behavioral responses in rodents [15,19,63].

### Expression of BET Proteins in the Brain

Though ubiquitously expressed in mammalian tissues, BRD2, BRD3, and BRD4 expression patterns in the CNS are not overlapping and vary depending on the area and neuronal subtype. Single-cell analysis showed that *Brd3* mRNA is expressed at high levels in most neuronal subtypes, while the amount of *Brd2* is the lowest. *Brd4* transcripts are especially abundant in striatal neurons expressing dopamine receptor 1 (D1R) and 2 (D2R) when compared to *Brd2* and *Brd3* [64,65]. In neurons, BRD4 activates gene transcription, suggesting that this epigenetic reader is an indispensable molecular link between neuronal activation and transcriptional responses [15]. Though incomplete, some evidence has shed new light into the molecular pathways linking BRD4 and neuronal activity (Figure 2). Indeed, neurotransmitter and neurotrophic factors induce signaling cascades leading to the activation of PKA and Casein Kinase 2 (CK2). These kinases, in turn, phosphorylate BRD4, favoring its binding to acetylated histones [15,45,66]. 

Another study documented elevated abundance of *Brd4* mRNA in rat striatal neurons, where BRD4 is involved in dopamine-induced and cAMP/PKA-dependent basal transcription. Indeed, cAMP/PKA signaling elicits BRD4 recruitment to genes induced by dopamine stimulation, whereas pharmacological or genetic inactivation of BRD4 significantly downregulates the transcription of a subset of genes mediated by D1R [66]. Consistent with the role of BRD4 in dopaminergic signaling, *Brd4* transcript levels are much higher than *Brd2* and *Brd3* in nucleus accumbens (NAc), a region enriched in D1R- and D2R-expressing neurons implicated in reward mechanisms [67]. This finding is further supported by the reportedly high concentration of BET proteins in the amygdala and midbrain, also involved in reward behavior in both rodents and non-human primates [52]. Other remarkable differences in the expression of specific BETs have been detected in some neuronal subtypes. For instance, *Brd2* expression is prevalent in cerebellar neurons, while the highest *Brd4* levels were found in claustral neurons of the frontal cortex and in some hippocampal neurons [65]. 

It is worth noting that the expression pattern of BET proteins in glial cells is more elusive and deserves further investigation. The few data available to date highlight BRD2, BRD3, and BRD4 expression in mouse astrocytes. However, lipopolysaccharide (LPS)-induced activation of astrocytes only enhances BRD2 levels, suggesting that other BET proteins may not be involved in astrogliosis elicited by inflammatory stimuli [69]. Similar to astrocytes, oligodendrocytes are also poorly characterized in terms of BET expression profile, even though a prominent expression of BRD2 over BRD3 and BRD4 was observed in this cell lineage [70]. The above-described peculiar pattern of expression of BET proteins may contribute to accomplish specific functions in a brain region- and cell type-dependent fashion. Despite this notion, future studies aimed at further characterizing the BET expression profile in the brain regions will certainly provide deeper insights into their functional implications in CNS pathophysiology. 

## 3. BET Proteins in Brain Physiology 

### 3.1. Neuronal Differentiation and Neurodevelopment

The process of neural differentiation during development is essential for the formation and proper functioning of the CNS. Neural stem cells (NSCs) give rise to cell populations progressively acquiring specific morphological and functional properties [71]. Among such features, elongation of axonal and dendritic processes guarantees synaptogenesis, axon guidance, and proper neuronal function [72]. Various transcription factors (TFs) and epigenetic modulators play key roles in neural cell commitment and differentiation [73,74]. In recent years, increasing evidence emphasized the central role of BETs in epigenetic regulation of neurodevelopment, being involved in multiple mechanisms underlying such processes. Li and colleagues demonstrated that inhibition of BET proteins by JQ1 promotes differentiation of mouse cortical neural progenitor cells (NPCs) towards a neuronal, rather than a glial, phenotype. In particular, genes related to cell cycle progression and glial differentiation were decreased, whereas pro-neurogenic gene expression was sustained. Consistently, BRD2, BRD3, and BRD4 levels decreased, indicating that their suppression is directly associated with neuronal differentiation [75].

To identify the regulatory mechanisms operating during embryonic neurogenesis, Westphal et al. (2020) observed that inhibiting BET proteins by JQ1 and I-BET151 interferes with dopaminergic and noradrenergic neurogenesis in zebrafish. Specifically, both molecules caused dramatic reduction in the expression of *th* gene, a marker for dopaminergic and noradrenergic differentiation, in the embryonic clusters of the telencephalon and pretectum. Furthermore, JQ1 strongly decreased the expression of *sox2*, a stem cell marker expressed in NPCs of the ventricular zone and of the retinal proliferation zone, suggesting a decrease in cell stemness following the pharmacological inhibition of BET proteins [76].

Among BET family members, BRD2 seems a crucial regulator strongly implicated in neurodevelopment, in relation to its wide expression in the developing forebrain, midbrain, hindbrain, and spinal cord [77]. *Brd2*-deficient mice showed early embryonic lethality, as well as generalized developmental and growth delays with impaired neurulation and consequent hindbrain exencephaly [53]. Embryos carrying a deletion in the Brd2 gene showed defects in neural tube closure at the cranial level and craniofacial malformations with irregular thickenings in the neuroepithelial layers of the rostral hindbrain [77]. These data indicate that BRD2 is required for proper neural tube closure, suggesting an essential role of this factor in regional specification of the developing hindbrain. RT-PCR and microarray analysis in *Brd2*−/− embryos revealed reduced expression of genes encoding for the transcriptional factors neurogenic differentiation factors 1 and 4 (*NeuroD1* and *NeuroD4*), suggesting a pivotal role of BRD2 in neuronal maturation [53]. Furthermore, BRD2 promotes cell cycle exit of neuroepithelial cells, and this event may depend, at least in part, on the E2F1-driven pathway. In fact, alterations in the process of neuronal differentiation and cell cycle progression have been found in *Brd2*-deficient neuroepithelial cells. Tuj1 (βIII-tubulin) expression and the number of Tuj1-positive neurons were severely reduced, demonstrating that neuronal differentiation was impaired. Surprisingly, these abnormalities were reversed through the suppression of *E2F1*, a gene encoding for a TF that binds BRD2 and controls cell cycle progression. These results suggested that BRD2 is required for cell cycle exit and for neuronal differentiation via the E2F1 pathway during the developing mouse CNS [77]. 

Although several lines of evidence reported the involvement of BRD2 in the cell cycle progression of NPCs, its expression has also been detected in differentiating neurons, including interneurons, motoneurons, and sensory neurons during neurodevelopment [78]. Despite its putative role in neuronal maturation, when overexpressed, BRD2 impairs differentiation, while increasing cyclin D1 and A2 levels [79]. Experimental findings identified the growth factor pleiotrophin (Ptn) as an important interactor of BRD2 in neuronal differentiation. Specifically, Ptn antagonizes BRD2 and its cell cycle stimulation, promoting neuronal differentiation in the neural tube of chick embryos and P19 cells. The researchers demonstrated that Ptn action is exerted through destabilization of BRD2 association to the chromatin. Consistently, *Ptn* knockdown reduced neuronal differentiation, and Ptn co-expression neutralized the effects of BRD2, favoring neuronal differentiation over cell proliferation [80].

Given the involvement of BET proteins in a plethora of processes that promote, maintain, and modulate neurodevelopment, it is not surprising that BRD4 has also been assigned to key functions in embryonic development. Recently, a novel role for BRD4 in the differentiation of mouse neural crest cells has been revealed. Indeed, embryos with neural crest-specific deletion of *Brd4* showed skeletal dysplasia, craniofacial defects, and cleft palate, as well as complex cardiac anomalies and consequent perinatal lethality [81]. Furthermore, *Brd4* knockout in mice leads to embryonic death in the early post-implantation stages [82].

During postnatal development, in the cerebellum, granule cell progenitors (GCPs) undergo symmetric division and exit the cell cycle, resulting in rapid cell expansion and differentiation [83]. Recently, Penas and coworkers illustrated that, after cell cycle exit, GCPs exhibit downregulation of BRD4 activity due to casein kinase 1δ (CK1δ)-dependent phosphorylation. Treatment with I-BET151 reduced the proliferation of GCPs in vitro; similarly, mice stimulated with the BET inhibitor JQ1 showed a reduction in cerebellar proliferation of GCPs. Consistently, *Brd4* knockout in the developing cerebellum resulted in aberrant cytoarchitecture, with the formation of the cerebellar layers that persisted into postnatal development; the reduced size of the cerebellum correlated with behavioral deficits in mice, which displayed symptoms of cerebellar ataxia. Taken together, these data demonstrate that BRD4 is an essential regulator of GCP proliferation and cerebellar development in vivo [84].

### 3.2. Cognitive Functions and Behavior

Healthy brain functioning requires fine control of gene expression in neural cells. Epigenetic regulatory mechanisms and chromatin remodeling are also crucial for cognitive functions such as learning and memory, which involve specific gene expression profiles that are necessary to ensure neural connectivity and plasticity [85,86].

BET proteins display high affinity for histone marks (H4K5/K8/K12ac) that are associated with learning and memory [65]. Additionally, BET inhibitors regulate the rapid activation of immediate early genes (IEGs) (e.g., *Bdnf*, *Nr4a1/2*, *Gria1*, *Fos*, *Arc*, *Egr1,* and *Junb*) whose expression is usually detected in cognitive processes [15,18,87]. BRD4 has been involved in the potentiated neuroplasticity and memory following histone deacetylase (HDAC) inhibition [87]. For example, HDAC3, the most highly expressed class I HDAC in the brain [88], is a negative regulator of different types of memory. HDCA3 inhibition resulted in enhanced novel object recognition (NOR), contextual fear conditioning, auditory memory encoding, and instrumental learning [89,90,91,92]. Pharmacological blockade of HDAC3 by RGFP966 increased subthreshold NOR memory in male C57BL/6 mice, while effects were reversed by cotreatment with the BET inhibitor JQ1. Particularly, during NOR training, no significant differences in exploration time were observed among experimental groups (control, RGRP966-treated, and RGFFP966/JQ1-cotreated mice). However, during the NOR test, time spent exploring a novel object was significantly increased by RGFP966, which was blocked by JQ1. Similarly, JQ1 administration counteracted the reward-related learning enhanced by RGFP966 [87]. In another study, mice treated with JQ1 showed no preference between a familiar object and a novel one in the NOR test. Strikingly, when mice were immediately tested after JQ1 administration, control and JQ1-treated mice performed equally well, indicating that JQ1 does not impact on learning or short-term memory, but instead impairs long-term memory. Furthermore, mice injected with a single dose of JQ1 after NOR training showed no preference for a novel object, suggesting that JQ1 can block long-term memory formation when administered during the process of memory consolidation. Korb and colleagues (2015) also tested a Pavlovian fear-conditioning protocol to assess the extent of memory deficits, demonstrating that mice treated with JQ1 were less able to discriminate between the training context and a new given context. This suggests that context discrimination dependent on the hippocampus may also involve BET protein activity [15]. The estimation of RNA levels of several memory-associated genes in the cortex and hippocampus isolated from fear-conditioned mice showed that JQ1 inhibited the induction of *Fsl2*, *Crem,* and *Bdnf* genes in at least one brain region. Consequently, the behavioral alterations caused by JQ1 during fear conditioning are likely related to the transcriptional modulation of key memory genes in vivo. Kim and colleagues (2021) also assessed the involvement of each BET family member in long-term memory formation in vivo. The expression of BRD2, BRD3, and BRD4 was selectively suppressed through gene silencing by stereotactic injection of adeno-associated viruses into the hippocampus. Two weeks later, mice were subjected to contextual fear conditioning. The effect of each knockdown on long-term memory was different: notably, the most severe impairments were observed upon BRD2 silencing, followed by knockdown of BRD4 and then BRD3 [93]. These results highlight that each BET family protein differentially contributes to long-term memory by regulating activity-dependent gene expression.

BET inhibition impairs the extinction of auditory fear memory but does not alter the acquisition of contextual fear conditioning, lithium chloride-conditioned place aversion, or Barner maze or Y-maze test. Fear extinction is commonly used to treat anxiety disorders such as post-traumatic stress disorder (PTSD) and specific phobias. During fear extinction, excess fear is suppressed by re-exposure to the fear-triggering stimulus in the absence of any aversive event [94,95,96].

Several studies showed that fear-related behavior is epigenetically regulated, even via histone acetylation [97,98]. For instance, fear extinction or conditioning could elicit histone H3 and H4 acetylation in the medial prefrontal cortex and hippocampus [99]. Consistent with the increased histone acetylation, Huang and colleagues (2021) observed that auditory fear conditioning in adult mice promoted a biphasic BRD4 activation in the anterior cingulate cortex (ACC) and hippocampus [100]. In this context, changes in IEG expression were thought to be indicative of specific activation of brain regions associated with fear responses. BRD4 levels in the ACC and hippocampus were also sustained two weeks after auditory fear conditioning, suggesting that neuronal structures in these regions may undergo continuous specific modifications which contribute to the extinction of remote fear memory. JQ1 administration ahead of fear conditioning failed to influence recent fear extinction, but it was able to impair remote fear extinction. Furthermore, Insulin-like growth factor 2 (IGF-2) was upregulated in the ACC following extinction of the remote fear memory, and JQ1 decreased the extinction level blocking IGF-2 upregulation [18]. Moreover, when JQ1 was administered 12 and 13 days after fear conditioning to suppress the second phase of BRD4 activation, fear extinction on the 14th day was impaired, suggesting that BRD4 activation after auditory fear conditioning in the ACC and hippocampus is related to remote fear extinction. JQ1 was sufficient to impair extinction of remote fear memory but to a lesser degree than *Brd4* conditional knockout in mice. *Brd4* KO mice exhibited an increase in fear response during both the extinction training and extinction test compared to control mice. However, JQ1-treated mice presented more fear-related behaviors (e.g., freezing) during the extinction test phase compared to the control group. The seeming discrepancy of these results may be explained by total abolishment of BRD4 activity in the knockout mouse vs. its only partial suppression following JQ1 treatment [100]. 

Effects on long-term memory and exploratory motor activity were reported in mice treated with JQ1 and the inhibitor I-BET858 [101]. However, other studies showed that different behaviors, such as locomotor activity, were not influenced by JQ1 [15,16,17,18,23,102], indicating that BET blockade does not cause widespread behavioral changes or alterations in motor function. 

The effects of BET inhibition on behavior are complex and occasionally contradictory among studies. Despite some studies suggesting that BET inhibition disrupts memory and induces autism-like behavioral deficits, others reported memory improvement in wild-type animals and the occurrence of beneficial effects in mouse models of neurological disorders [66]. To deeply understand the nuances of BET function, it will be necessary to elucidate the mechanisms of BET regulation in different brain regions and in response to various neuronal signaling pathways.

In most reports, effects on learning and memory were assessed by intraperitoneal administration of BET inhibitors [15,16,23,67,102]. Even though JQ1 and derivatives employed in these studies are able to cross the BBB, it is unclear whether BET inhibitors readily reach all the brain regions involved in cognition and behavior at adequate concentrations [65]. Additionally, experimental evidence collected so far is based on the use of general BET inhibitors unable to either act on specific BET proteins or discriminate between bromodomains within the same BET molecule. Further research using genetic manipulation or more selective BET inhibitors may be required to accurately unveil the role exerted by BET proteins in specific aspects of cognition.

## 4. Involvement of BET Proteins in Neuropathological Conditions

### 4.1. Neurodevelopmental Disorders

Given the complex and delicate regulation ensuring a proper CNS morphogenesis and differentiation, diverse genetic and environmental factors can determine the onset of neurodevelopmental disorders, hallmarked by alterations in cognitive and motor functions. Xiang and colleagues (2020) recently identified BRD4 as a critical factor in Rett syndrome (RTT), an X-linked neurodevelopmental disorder caused, in most cases, by loss of function in methyl-CpG binding protein 2 (MeCP2). An abnormal increase in BRD4 chromatin binding was found in MeCP2-mutated human cortical interneurons, and JQ1 treatment attenuated transcriptional hyperactivation by decreasing BRD4 engagement to chromatin. Furthermore, human median ganglionic eminence organoids (hMGEs) with MeCP2 mutation displayed altered synchronization of calcium spikes, which was restored after JQ1 treatment. Similarly, different neuronal TFs such as MEF2C and NEUROD2 were upregulated but were recovered upon JQ1 treatment in MeCP2-mutated neurons derived from human cortical organoids (hCOs). Surprisingly, pharmacological inhibition of BET proteins by JQ1 in the MeCP2/Y mouse model resulted in remarkably extended lifespan (about 81% vs. untreated MeCP2/Y mice). Furthermore, while MeCP2/Y mice showed pathological progression typical of RTT, treatment with JQ1 significantly reduced symptoms in MeCP2/Y mice [103]. Taken together, these data suggest that BRD4 overactivation may impair neuronal development, as it may result in dysregulation of genes related to neuronal differentiation and activity, and that epigenetic regulation exerted by BRD4 is essential for optimal neuronal function.

Similarly, in a mouse model of fragile X syndrome (FXS), BRD4 blockade alleviated the transcriptional dysfunction and the behavioral phenotypes associated with the disease [102]. FXS is a monogenic neurodevelopmental disorder resulting from the loss of function of fragile X mental retardation protein (FMRP), codified by *Fmr1* gene, which represses the translation of target transcripts and is particularly implicated in the regulation of synaptic function and plasticity [104,105,106]. In mice, BRD4 expression decreases during neuronal maturation but remains expressed in the adult stage [15]. Conversely, although BRD4 expression is reduced in Fmr1 knockout (KO) neurons during development, a higher expression is already present in the early stages of development compared to WT. In addition, WT neurons display lower levels of BRD4 phosphorylation than Fmr1 KO neurons. Consistently, elevated BRD4 expression was also found in Fmr1 KO young adult brains. Pharmacological inhibition by JQ1 resulted in a decreased expression of genes upregulated in Fmr1 KO neurons and in a significant rescue of synaptic spine number. Moreover, it was found that Fmr1 KO mice were more prone to compulsive- and repetitive-like behaviors when compared to control WT mice, whereas JQ1 treatment restored this phenotype. As to social behaviors, KO Fmr1 mice showed abnormal social interaction, which was reversed by JQ1 treatment, without affecting overall movement and exploratory activity. Different from Frm1 KO mice, no significant behavioral changes were reported in WT mice treated with JQ1. These findings suggest that JQ1 administration may rescue physiologically relevant aspects of neuronal function in mice with an FXS phenotype, characterized by overactive transcription and neuronal hyperexcitability. In contrast, JQ1 has a deleterious effect in WT mice, blocking memory formation. The effects of JQ1 in FXS were further evaluated by testing the co-administration of CX-4945. The concurrent CK2 inhibition by CX-4945 efficiently attenuated the abnormalities in social interaction, indicating that targeting BRD4 or its phosphorylation status is a promising approach to reverse some of the cognitive deficits observed in FXS [102].

This fact corroborates the finding that pharmacological suppression of BET proteins during mice adolescence leads to selective repression of neuronal gene expression and development of autism-like syndrome, supporting the model in which proper amounts of BRD4 are required for optimal neuronal function [19,102,107].

In line with these data, BET protein expression was found to be altered in cortices isolated from WT mice at postnatal day (P) 20, compared to Fmr1 KO mice. Specifically, Fmr1 KO mice showed increased BRD4 levels and reduced BRD2/3 levels compared with WT. ChIP-seq analysis for each BET family member in cortical tissues isolated at P60 demonstrated that the binding of BRD2 and BRD3 to chromatin cis-regulatory regions was significantly reduced in Fmr1 KO mice, whereas BRD4 recruitment was unchanged, except for the enhancer regions. These results suggest that alterations in the recruitment of different BET proteins to their respective regulatory regions could contribute to the transcriptional abnormality observed in FXS. Next, the role of CBP/p300 HAT was examined, as its acetylating activity provides the substrates for the binding of BET proteins to chromatin. Unexpectedly, pharmacological inhibition of CBP/p300 HAT by C646 enhanced the recruitment of BRD2 to cis-regulatory regions, whereas no change was observed for BRD3. On the other hand, BRD4 was the only member of the BET family that showed decreased recruitment to chromatin following C646 treatment. Specifically, increased BRD2 binding was noted in regions where BRD4 binding was decreased, suggesting a compensatory mechanism. In addition to the enzymatic activity, CBP/p300 is an important coactivator implicated in the protein–protein interactions of various transcriptional regulators. Therefore, the authors investigated whether BET proteins require the coactivator function of CBP/p300 for their recruitment. CBP knockdown resulted in decreased binding of BET proteins to promoters and enhancers, suggesting that BET recruitment to regulatory regions depends, at least in part, on CBP coactivation [93]. Collectively, these data suggest that BET family members do not simply act in functional redundancy, but efficiently coordinate with each other, and that impaired BET coordination is implicated in the altered transcription and subsequent pathological conditions manifested in FXS.

Increasing evidence also highlights the putative involvement of BET proteins in Cornelia de Lange syndrome (CdLS), a rare developmental multisystemic disorder characterized by cognitive impairment and altered physical and behavioral features. CdLS is part of the “cohesinopathies” family of developmental disorders, associated with mutations in proteins functionally related to the regulation of chromatin folding. Indeed, approximately 70% of CdLS cases present with mutations in the cohesin-loading factor NIPBL, which promotes cohesins association to DNA [108]. It is worth noting that CdLS-like phenotypes were recently associated with mutations in BRD4 [109,110,111] and, intriguingly, BRD4 was shown to interact with NIPBL [109,112], cooperating in the transcriptional regulation of several genes involved in development [109,110]. Heterozygous mice for *Brd4* (*Brd4^+/−^*) and *Nipbl* (*Nipbl*^+/−^) genes showed similar phenotypes: animals that survived the perinatal period exhibited significantly smaller size, craniofacial abnormalities, and brain alterations [82,113]. Olley and collaborators observed that a de novo mutation in the second BrD of BRD4 impairs its recognition of acetylated histones on chromatin, though not affecting its interaction with NIPBL [109]. Subsequently, it was demonstrated that the recognition and interaction pattern for NIPBL is the ETD of BRD4, and proteins stabilize each other on chromatin by modulating the expression of several developmental genes in both mice and humans. Consistently, the expression of a truncated version of BRD4 lacking ETD led to the dissociation of NIPBL from chromatin, suggesting that NIPBL activity might partially depend on cooperation with BRD4 [112]. In agreement, BRD4 depletion caused a reduction in chromatin occupancy by NIPBL, resulting in loss of normal genome folding in murine embryonic stem cells (mESCs). The binding is restored by exogenously delivering the full-length BRD4, but not BRD4 with ETD deletion, confirming that the ETD of BRD4 is the recognition pattern required for NIPBL interaction. Interestingly, point mutations in the ETD of BRD4 that prevented binding to NIPBL impeded in vitro differentiation of neural crest cells into smooth muscle cells [81].

A plethora of studies have demonstrated the indisputable physiological role of BET proteins as crucial epigenetic regulators in neurodevelopment and in neuronal functions. Indeed, it is not surprising that an altered function of BET proteins is directly or indirectly implicated in neurodevelopmental disorders. Therefore, targeting BET proteins could provide new and effective treatments for different pathological conditions characterized by impairments in their function. 

### 4.2. Neuroinflammation

Neuroinflammation is a significant driver in the pathogenesis and progression of several neurological conditions [65,114,115,116,117]. A growing body of evidence highlights a pivotal role of epigenetic signaling in the modulation of inflammation. During the last decade, the involvement of BET proteins in this process has been characterized by means of chemical inhibitors or genetic approaches. The first link between BET and inflammation was established in the in vivo study by Nicodeme and colleagues, showing that BET blockade suppressed the synthesis of pro-inflammatory mediators in activated macrophages and protected mice from lethal shock induced by LPS administration [12]. Other studies confirmed these findings, suggesting that BETs are essential in orchestrating the inflammatory response [12,118,119,120,121]. Such modulation operated by BET proteins depends on their interaction with NF-kB, a crucial transcription factor in inflammation. In detail, the p65/c-Rel subunit of NF-kB is acetylated at the Lys310 level by p300/CBP, thus allowing the recruitment of BRD4 near NF-kB target genes [122]. Accordingly, BET inhibition attenuates the expression of the main inflammatory genes and, in turn, induces an anti-inflammatory response in a variety of tissues and organs, including the brain (Figure 3).

In the nervous tissue, microglia are key players in the regulation of innate immune response and neuroinflammation [125,126]. Specifically, recent data demonstrate that enhanced degradation of BET proteins achieved by PROteolysis-TArgeting Chimera (PROTAC) technology blunted the LPS-induced pro-inflammatory response in the murine microglial cell line SIM-A9. In detail, treatment with dBET1 induced a significant degradation of BRD2 and BRD4, which was associated with a considerable reduction in iNOS and COX-2 levels. Furthermore, BET degradation attenuated the expression of different pro-inflammatory genes, such as *Nos2*, *Ptgs2*, *Il-1β*, *Tnfα*, *Ccl2*, *Il-6,* and *Mmp9*. The effect of BET inhibition on the transcription of inflammatory genes was confirmed by evaluating the protein expression of these pro-inflammatory modulators upon JQ1 treatment [127]. Other data demonstrated that JQ1 stimulation of LPS-treated human microglial clone 3 (HMC3) cells not only exerts anti-inflammatory effects, but also anti-migratory activities [57]. 

Migration of activated microglial cells towards damaged sites is strongly associated with the inflammatory response [128,129] and this is ensured by the expression of migration-related genes such as *Mmp3*, *Mmp13*, *Csf2*, *Ido1*, *Tnfsf10,* and *Vcam1*. qRT-PCR analysis showed a reduced expression of migration-related genes in HMC3 cells treated with JQ1. Pharmacological BET inhibition also determined BRD4 displacement from the promoter of *Irf1* gene, coding for a TF involved in microglia activation [57]. Consistently, JQ1 administration counteracted LPS-mediated inflammatory response in BV2 microglial cells. In this context, BET blockage interfered with NF-kB activation by impeding the nuclear translocation of the p65 subunit. Precisely, JQ1 treatment blocked the activating phosphorylation of IKKα/β, a kinase complex essential for the induction of NF-kB cascade, whose activation was particularly pronounced upon LPS stimulation [123]. In addition, JQ1 also inhibited the MAPK axis, another signaling cascade responsible for NF-kB-mediated transcriptional activity (Figure 3). Therefore, these results revealed that BET proteins regulate microglia activity at multiple levels, affecting the activation of diverse pathways involved in the production of pro-inflammatory mediators and cell migration.

Besides microglia, astrocytes also contribute to the inflammatory response in the CNS. Treatment with LPS promoted a significant increase in *Brd2* mRNA levels on primary murine astrocytes [69]. BET inhibition suppressed mRNA levels of pro-inflammatory cytokines [130]. Further confirmation of the involvement of BET proteins in astrocyte- and microglia-mediated neuroinflammation was provided from studies conducted on GFP-IL-1β transgenic mice. Intraperitoneal administration of LPS induced microgliosis and astrogliosis, whereas BET inhibition by JQ1 markedly reduced astrocyte and microglial activation, as assessed by the expression levels of CD68 and GFAP, as well as the levels of pro-inflammatory cytokines IL-1β, IL-6, and TNF-α. Furthermore, in vivo studies demonstrated that, similar to the results obtained on BV2 microglial cell cultures, the activation of the MAPK/NF-kB signaling cascade mediated by LPS in brain tissue is significantly attenuated by JQ1 [123].

Recent studies employed models of permanent and transient cerebral ischemia to evaluate the effects of BET inhibition on the inflammatory response mediated by NF-kB [131,132,133,134]. In rats, transient cerebral ischemia obtained by middle cerebral artery occlusion (MCAO) resulted in a significant increase in BRD4 expression in the MCAO group compared to the control group, which was prevented by JQ1 treatment. JQ1 administration also hindered the infarct volume, reduced the number of apoptotic cells, and decreased the expression of pro-inflammatory factors (IL-1β, IL-6, IL-17, and TNF-α) in the ischemic brain. Notably, BET blockade suppressed neuroinflammation by reducing p65 levels and by increasing the cytosolic expression of the NF-kB inhibitor IkB [131]. Other reports corroborate these findings, demonstrating that NF-kB suppression by BET blockade markedly hampered the expression of pro-inflammatory mediators, leading to the reduction in pyroptosis and inflammasome activation. From a functional point of view, the molecular changes promoted by BET inhibition were associated with a partial recovery of neurological deficits [132]. Neuroprotection mediated by BET inhibition was also confirmed in mouse models of permanent cerebral ischemia. For instance, BRD4 blockade by using dBET1, a proteolysis-targeting chimera, attenuated the infarct volume in permanent focal cerebral ischemia, and this was related to a decrease in pro-inflammatory mediators such as TNF-α, CXCL1, CXCL10, CCL2, and matrix metalloproteinase-9 (MMP-9). dBET1 administration significantly prevented BBB abnormalities, as well as neutrophil infiltration into the ischemic area. Interestingly, dBET1 significantly ameliorated the neurological symptomatology [134]. Reduction in inflammation and preserved BBB integrity following BET inhibition was also reported in the C57BL/6J mouse model subjected to transient MCAO. Indeed, dBET1 reduced pro-inflammatory cytokines, neutrophil infiltration, MMP-9 protein levels, and infarct volume. The protective effects of dBET1 against ischemic damage are attributable not only to its anti-inflammatory function, but also to its anti-oxidant function. In fact, evaluation of oxidative stress in MCAO mice revealed an increase in the oxidative damage marker 4-hydroxy-2-nonenal (4-HNE), a concurrent buildup in the protein levels of GP91phox (NOX2) subunit of the pro-oxidant NADPH oxidase complex, and a decrease in anti-oxidant enzymes such as superoxide dismutase 2 (SOD2) and glutathione peroxidase 1 (GPX1). On the contrary, dBET1 significantly attenuated the levels of 4-HNE and NOX2 and simultaneously increased SOD2 and GPX1 expression [133]. These effects could be attributable to the ability of BET proteins to modulate Nrf2-dependent transcription of anti-oxidant genes. Several studies highlighted that BET proteins negatively affected Nrf2 signaling, which was prevented by the administration of BET inhibitors [135]. Furthermore, under specific physiopathological conditions, BRD2 and BRD4 directly bind to the promoter of *Nox2*, *Nox4*, *p47phox,* and *p67phox* genes. Coherently, JQ1 treatment promoted the displacement of BET proteins from the chromatin, reducing the transcription of NADPH oxidase subunits and the subsequent induction of oxidative stress [136]. Taken together, this evidence suggests that BET modulation could affect redox balance by influencing Nrf2 and NADPH oxidase activity in ischemic conditions.

An additional function attributable to BET proteins in stroke-associated neuroinflammation has emerged from a recent study focusing the attention on the formation of fibrotic scars. It has been shown that TGF-β1 stimulation increased BRD4 expression in a cell culture model of fibrosis and simultaneously promoted fibroblast proliferation and migration. In contrast, JQ1 blocked these effects. The results obtained from in vitro studies were further supported by in vivo models. Specifically, rats underwent transient MCAO developed fibrosis in the infarcted area, which was attenuated when BRD4 was knocked down by adenovirus. Concurrently, ischemic damage, infarct volume, and cognitive alterations appeared to be reduced [137]. 

Recent evidence underlines that BET inhibition may also have a role in the neuroinflammatory conditions associated with sepsis-induced encephalopathy (SAE). Indeed, JQ1 treatment blocked NF-kB signaling and decreased inflammosome activation in the hippocampus of an experimental mouse model of SAE, leading to the suppression of the canonical pyroptosis pathway and the release of pro-inflammatory factors. Furthermore, BET inhibition selectively suppressed hippocampal microglia activation in SAE mice, determining an overall protection against BBB breakdown and neuronal damage [138]. 

Thus, it is becoming increasingly clear that BET proteins play an essential role in the modulation of complex neuroinflammatory pathways and that they are functionally connected to numerous CNS pathologies characterized by oxinflammation.

### 4.3. Neurodegenerative Diseases

Neurodegenerative diseases such as AD, Parkinson’s disease (PD), Huntington disease (HD), amyotrophic lateral sclerosis (ALS), or frontotemporal dementia (FTD) are characterized by a progressive decline in neuronal homeostasis which eventually leads to cell death. Although the etiological mechanisms at the root of these pathologies are different, they share similar degenerative aspects affecting neuronal and non-neuronal cells: notably, changes in gene expression, protein aggregation, redox disbalance, and neuroinflammation are common hallmarks of these conditions [139,140]. As mentioned in the previous sections, regulation of gene transcription can be achieved through epigenetic changes concerning histone tail modifications such as acetylation and methylation [141]. Targeting these modifications has shown promising results in the treatment for neurodegenerative diseases, especially concerning histone writers and erasers such as HAT and HDAC, respectively, and readers of histone acetylome such as the BET proteins (Table 1). 

For instance, emerging evidence suggests that BET inhibitors could be valuable targets to treat AD and other neurodegenerative diseases [65,142]. Experimental findings demonstrated the beneficial effects of BET blockade in both wild type and AD mouse models; JQ1 treatment showed an improvement in spatial and associative memory, as well as in long-term potentiation (LTP). RNA-seq analysis revealed that these functional data were associated with the upregulation of genes involved in ion channels’ activity and DNA repair system [23]. Another in vivo study conducted on male adult Wistar rats mimicking AD pathology supported previous studies. JQ1 treatment significantly ameliorated spatial memory acquisition and retrieval, enhanced the phosphorylation of CREB, a critical transcription factor for memory consolidation, and increased the expression of the synaptic markers postsynaptic density 95 protein (PSD95) and synaptophysin. Concurrently, the proinflammatory mediator TNF-α was significantly reduced upon BET blockade. Interestingly, it has been observed that the administration of fluorocitrate (FC), an inhibitor of astrocyte metabolism, prevented the JQ1-induced phenotypic amelioration by inducing the downregulation of p-CREB, PSD95, and synaptophysin. Co-administration of FC also led to a buildup of TNF-α. Even though the molecular link between FC and TNF-α elevation needs to be clarified, FC may suppress the neuroprotective activity of astrocytes, thus accelerating neuronal dysfunction and exacerbating cognitive alteration, regardless of the presence of JQ1 [143]. In addition, previous studies highlighted that proinflammatory microglia could trigger the polarization of astrocytes into a neurotoxic phenotype [144]. Since the authors suggested that microglia could represent the source responsible for TNF-α upregulation [143], it is possible to speculate that the interplay between microglia and astrocytes may cause the nullifying effects of FC on memory improvement mediated by JQ1. The beneficial effect of BET inhibition against the neuroinflammatory condition in AD has been further supported by Magistri and colleagues; they found that JQ1 exhibited decreased mRNA expression of pro-inflammatory cytokines and chemokines such as *Il-1b*, *Il-6*, *Nos2*, *Tnfa,* and *Ccl2* in the 3xTg mouse model of AD. However, different from other published evidence, they failed in detecting cognitive improvement; this discrepancy can be explained by divergences in the experimental setup, such as age of the animals, dosage used for treatments, animal model training, and behavioral assessment [17]. The relevance of BET proteins to neuroinflammation in AD was sustained by another study showing that BET blockade reduced phagocytic activity of the microglial cell line BV2; the effect was dependent on the downregulation of phagocytosis-related genes which were involved in the pathogenetic mechanisms of AD [145]. The pharmacological potential of BET inhibition in AD has also been investigated in co-treatment with HDAC inhibitors. Male rats were treated with JQ1 and/or MS-275 to inhibit BET or HDAC proteins, respectively. Monotherapies, as well as combined therapies, were effective in counteracting cognitive impairments, CREB suppression, and elevation of TNF-α induced by Aβ administration. Notably, combined therapy did not show any synergic effect [56]. Although most of the literature points out a neuroprotective role of BET inhibition in AD, a very recent report demonstrated that inhibition or degradation of BET protein BRD4, obtained by treatment with JQ1 and ARV-825, respectively, enhanced Tau hyperphosphorylation and Aβ levels as a result of beta-site amyloid precursor protein cleaving enzyme 1 (BACE1) activation in H4-APP751 human neuroblastoma and 3D-AD human neural cell culture. Indeed, both compounds were capable of augmenting p-Tau levels and the cleavage of the soluble amyloid precursor protein (sAPP) into Aβ through BACE1 pathways. However, *BACE1* mRNA did not display any increase after drug treatment, suggesting a possible post-translational mechanism of activation [146]. Collectively, the available data suggest a possible role of BET proteins in the pathogenic mechanisms of AD. However, further research is needed to better clarify their contribution and the effective employment of BET inhibitors as a putative pharmacological approach. 

Recently, apabetalone (RVX-208), a small molecule BET inhibitor, has been assessed for secondary prevention of cardiovascular disease (CVD) events in a randomized, placebo-controlled clinical trial (BETonMACE) with 2425 patients with diabetes and acute coronary syndrome [147]. Besides the favorable trend in the incidence of major cardiovascular events (MACE), Cummings and colleagues also explored the effects of apabetalone on cognitive function in this population with risk factors for AD. In this sub-study, cognitive functions of BETonMACE participants of 70 or more years of age were collected using the Montreal Cognitive Assessment (MoCA) score. Apabetalone treatment was associated with improved cognition as measured by MoCA scores, highlighting a novel therapeutic approach based on BET inhibitors for patients with concurrent CVD and cognitive impairment [148]. 

**Table 1 biomedicines-11-00750-t001:** Effects of BET inhibition in neurodegenerative disorders. ↑ indicates an increase; ↓ indicates a decrease. AD (Alzheimer’s disease); PD (Parkinson’s disease); HD (Huntington’s disease); ALS (Amyotrophic Lateral Sclerosis); FTD (FrontoTemporal Dementia).

Disease	Experimental Model	BET Inhibition	Pathways/Processes	Functional Effects	References
AD	APP/PS1-21 mouse	− JQ1	− LTP − DNA repair − Ions homeostasis	− ↑ Spatial and associative memories − ↑ DNA repair − ↑ Ion channel activity	[23]
	3xTg mouse	− JQ1	− Neuroinflammation	− ↓ Neuroinflammation (TNF-α, IL-1β, IL-6, Nos2, Ccl2); − No effects on cognition	[17]
	Wistar rats	− JQ1	− Synaptic plasticity − Neuroinflammation	− ↑ CREB signaling − ↑ Synaptic proteins − ↑ Spatial memory − ↓ Neuroinflammation	[143]
	Wistar rats	− JQ1	− CREB signaling − TNF-α signaling	− ↑ Spatial and aversive memories − ↓ Neuroinflammation	[56]
	− H4-APP751 cells − 3D-AD cells	− JQ1 − ARV-825	− Tauopathy − APP metabolism	− ↑ p-Tau aggregates − ↑ APP processing (↑ BACE1, ↑ Aβ formation)	[146]
	BV2 murine microglial cells	− JQ1 − *shBRD2* − *shBRD3* − *shBRD4*	− Expression of phagocytosis-related genes	− ↓ microglial phagocytic activity	[145]
	Clinical study (NCT02586155)	Apabetalone (RVX-208)	− Cognitive performances	− ↑ Cognitive function	[148]
PD	6-OHDA rat model	− JQ1	− Gene expression	− ↓ LID-associated immediate-early genes − ↓ LID-associated symptoms	[149]
	6-OHDA rat model	− JQ1	− Neuroinflammation	− Inhibition of neuroinflammation (↓ TNF-α, ↓ iNOS, ↓ IL-1β, ↓ IL-6, ↓ CD68, ↓ GFAP)	[150]
HD	− R6/2 mouse	− JQ1	− Gene transcription − mHTT aggregation	− ↑ Insoluble mHTT − ↑ Weight loss − ↑ Behavioral impairments	[151]
ALS/ FTD	Cells derived from ALS patients	− JQ1 − I-BET762 − I-BET151	− *C9ORF72* expression	− ↑ *C9ORF72* expression	[152]
	− SH-SY5Y G4C2 cell line − C9BAC mouse	− PFI-1 − JQ1 − OTX-015	− *C9ORF72* transcription − RNA *foci* − DPR inclusions	− ↑ V1-V3 *C9ORF72* transcription; − Increased RNA *foci*; − ↓ Toxic DPR inclusions; − ↓ Hippocampal-dependent cognitive deficits	[153]

Severe cognitive impairment can also represent a long-term complication of diabetes, characterized by reduced mental flexibility, intelligence, and speed of information processing [154]. Using streptozotocin-induced diabetic rats, Liang and colleagues performed a Morris water maze test, demonstrating that cognitive function in diabetic rats is deeply impaired but abrogated by JQ1 treatment [155]. 

Besides the role of BET proteins in AD, emerging studies imply the activity of these proteins in other neurodegenerative disorders. As a matter of fact, the inhibition of BET proteins in PD recently received attention for its potential effect in mitigating the side effects associated with drug treatments. Currently, Levodopa (L-DOPA), the precursor of dopamine, is the most efficient therapeutic approach for PD in clinical practice; unfortunately, its long-term use favors the appearance of drug-related adverse events, such as L-DOPA-induced dyskinesia (LID) [156]. Figge and colleagues examined the inhibition of BET proteins using a 6-hydroxydopamine (6-OHDA) rodent model of LID. In this context, BET protein expression was shown to be dysregulated, as well as the occupancy at promoter and enhancer regions of genes involved in dyskinesia development. Accordingly, treatment with JQ1 prevented LID and blocked the transcription of the immediate-early genes involved in the onset of dyskinesia [149]. Given that CK2 has been proven to be pivotal in BRD4 genomic localization [45] and since CK2 dysregulation has been previously associated with the onset of LID [157,158], it is possible to hypothesize that blocking BRD4 prevents its CK2-dependent delocalization, resulting in LID decline. In support of this evidence, another study better dissected the role of JQ1 in LID in a 6-OHDA rat model, showing that JQ1 alleviated LID by inhibiting neuroinflammation without affecting motor amelioration. In fact, treatment with L-DOPA promoted the expression of pro-inflammatory cytokines such as TNF-α, iNOS, Il-1β, Il-6, and glial activation marker CD68 and GFAP, which were in turn downregulated after JQ1 treatment. Hence, BET protein inhibition strongly blocked canonical NF-kB activation in the *striatum*, mitigating neuroinflammation and LID-associated symptoms [150]. Despite these findings, the scientific literature is still limited, and further studies are needed to better comprehend the involvement of BET inhibition in the physiopathology of PD.

HD is another neurodegeneration that could be influenced by the activity of BET proteins. HD is a genetic disorder caused by a mutation in *Htt* gene encoding for a multifunctional protein. The transcriptional dysregulation occurs at the early stage of the pathology, determining progressive neurodegeneration (firstly in the *striatum* and subsequently in the cortex) [159,160]. Concerning epigenetic changes, it has been postulated that *Htt* mutation severely affects HAT enzymatic activity; for this reason, it is believed that alterations in HAT functionality may represent a crucial phenomenon in transcriptional deregulation observed in HD [161]. In 2003, a research article elucidated the effects of HDAC on HD phenotype, reporting a positive impact in an R6/2 HD murine model treated with suberoylanilide hydroxamic acid (SAHA), a well-known HDAC inhibitor. In fact, mice showed a consistent improvement in a rotarod test, indicating a significant amelioration in motor activity when compared to the control group, without exhibiting a decrease in poliQ aggregation [162]. Similar experiments were conducted on an HD-N171-82Q transgenic mice model; the HDAC inhibitor phenylbutyrate decreased *striatum* degeneration and prolonged the survival rate. Contrary to the beneficial effects obtained with HDAC inhibitors, discouraging results were achieved by using BET inhibitors. Notably, it has been demonstrated that JQ1 improved motor skills and promoted the expression of genes involved in energy metabolism and protein translation in non-transgenic mice (NT). On the contrary, no significant effects were observed on R6/2 mice in a rotarod test. In addition, BET suppression was detrimental for pole test performance and aggravated HD-associated weight loss. Molecular analysis of R6/2 brain tissues displayed that JQ1 exacerbated HTT accumulation and increased the expression of genes related to the immune response and apoptosis, leading to a concurrent downregulation of genes associated with ion channel activity and functioning [151]. Studies on the effect of BET inhibition in HD models are still very limited; it would certainly be advantageous to validate the effects on different animal models or at different pharmacological dosages. 

Pharmacologically active molecules targeting the epigenome have been extensively studied in the context of ALS. About 10% of ALS cases are characterized by mutations in genes encoding for *SOD1*, transactive response DNA-binding protein 43 (*TARDBP*), and *C9ORF72*. Pathogenic mutations in *TARDBP* and *C9ORF72* are also involved in the onset of FTD; despite being very different clinically, patients with ALS and FTD share histopathological redundancy (classified as C9ALS/FTD) [163,164,165,166]. There are three substantial pathogenic mechanisms concerning this condition: (a) repeat-associated non-ATG (RAN) of *C9ORF72* mRNA leads to the accumulation of toxic dipeptide repeats (DPR) proteins, which generate neuronal inclusions [167]; (b) bidirectional *C9ORF72* mutated gene transcription forms RNA *foci*, whose toxic or neuroprotective effects still remain to be elucidated [168,169]; (c) *C9ORF72* loss of function or transcriptional impairment based on epigenetic modification (e.g., lysine 9 and 27 trimethylation of H3; DNA methylation on *C9ORF72* promoter; hexanucleotide repeat sequence) [170,171,172,173]. Novel pharmacological strategies are aimed at upregulating *C9ORF72* expression, since it has been recently proved that low expression levels impair autophagy, affecting the clearance of toxic DPRs [167,174]. Zeier and colleagues provided solid evidence that BET inhibitors increased *C9ORF72* expression in C9/ALS motor neurons [152]. Supporting data were recently obtained by Quezada and collaborators, showing that the BET inhibitor PFI-1 induced V1-V3 transcripts of the mutant *C9ORF72* gene and facilitated the formation of nuclear RNA *foci* with a consistent reduction in DPR inclusions in cell models of C9ALS/FTD. In addition, BET blockade suppressed the hippocampal-dependent cognitive deficits in a C9BAC mouse model of C9ALS/FTD [153]. Hence, BET inhibitors could be considered as a valuable therapeutic approach for ALS and FTD, even though further studies are essential to better identify the effective dosage, as well as the long-term efficacy in the pathology progression. Alongside a deeper molecular characterization of the phenotypical deficits on cell models, other tests should be run on animal models to assess cognitive improvement with further behavioral evaluation before approaching clinical trials.

### 4.4. Neuropsychiatric Disorders

Several psychiatric disorders are associated with alterations in histone modifications [175,176,177,178,179,180]. Chronic drug consumption is known to alter histone acetylation in brain regions associated with reward behaviors. Hence, epigenetic pharmacotherapies have emerged as a promising treatment approach for substance use disorder (SUD) due to their ability to reverse the maladaptive behavioral responses to drugs of abuse. Particularly, pharmacological inhibition of BET proteins has been shown to normalize the behavioral symptoms in a wide range of disease models, including SUD [65,181] (Table 2). In preclinical SUD studies, JQ1 was able to ameliorate the behavioral responses to different types of substances of abuse such cocaine, heroin, amphetamine, and nicotine in mice and rats [63,67,182].

Chronic cocaine administration results in a general increase in histone acetylation in brain areas related to reward, such as the NAc [183]. Additionally, manipulation of HAT and HDAC dramatically influences behavioral and molecular responses to psychostimulants in rodents [176,177,178,183,184,185,186,187,188,189,190]. Since histone acetylation influences drug-induced neuroadaptations and behaviors, the involvement of epigenetic readers, such as BET proteins, has been recently assessed in animal models of SUD. BRD4 protein levels, but not the expression of BRD2 and BRD3, was significantly increased in the NAc of mice and rats following repeated cocaine injections and self-administration. Through the conditioned place preference (CPP), a procedure utilized to study the role of context associations in reward-related behaviors, systemic and intra-accumbal administration of JQ1 was found to reduce cocaine CPP, without affecting locomotor activity or other types of learning [16]. Studying the underlying mechanisms, it has been revealed that repeated cocaine administration increases BRD4 binding to the *gria2* and *bdnf* promoter regions in the NAc, indicating that CK2-mediated phosphorylation of BRD4 is essential for cocaine addiction and relapse. Conversely, the inhibition of CK2-induced phosphorylation markedly represses cocaine effects. These findings indicate that BRD4 phosphorylation is required in long-term neuroplasticity and relapse associated with cocaine consumption [67]. Retrieval, reconsolidation of drug memories, and cocaine-induced reinstatement of drug-seeking behaviors find their molecular basis in the enhanced AMPA-mediated glutamate transmission in the NAc [67,191]. Several drugs targeting the glutamatergic circuitry in the NAc have been investigated in preclinical and/or clinical studies to alleviate drug relapse. For example, it was found that JQ1 hampered the GluA2 and BDNF expression induced by cocaine exposure. Coherently, the CK2 inhibitor CX-4945 was capable of altering cocaine conditioning, extinction, and reinstatement, demonstrating that CK2 inhibition may counteract the adaptive processes involved in cocaine-seeking behaviors.

More recently, the role of BET proteins in conditioned behaviors to other drugs of abuse, such as nicotine, amphetamine, morphine, and oxycodone, was investigated. These studies revealed that JQ1 administration attenuated nicotine and amphetamine CPP but did not alter morphine or oxycodone CPP. Importantly, several BET target genes that are known to regulate amphetamine- and nicotine-seeking behaviors were reduced upon the administration of JQ1 or other BET inhibitors. 

**Table 2 biomedicines-11-00750-t002:** Effects of BET inhibition in neuropsychiatric disorders. ↑ indicates an increase; ↓ indicates a decrease. SUD (Substance Use Disorder); PTSD (Post-Traumatic Stress Disorder); SZ (Schizophrenia).

Disease	Experimental Model	BET Inhibition	Pathways/Processes	Functional Effects	References
SUD	− C57BL/6 mice − Sprague Dawley rats	− JQ1	− Cocaine-induced neuroplasticity	− ↑ Behavioral response to drug addiction	[16]
	Long Evans rats	− JQ1	− Opioid-induced − chromatin conformation	− ↓ Heroin self-administration − ↓ Cue-induced drug-seeking behavior	[182]
PTSD	C57BL/6 mice	− JQ1	− Neuroplasticity associated with remote fear memory	− ↑ Behavioral impairment − ↓ IGF-2 upregulation	[18]
	C57BL/6 mice	− JQ1	− Auditory fear-conditioned memory	− ↓ BRD4 fear-induced activation − ↓ Remote fear memory extinction	[100]
SZ	− Neurons from SZ patients − WT mice	− JQ1	− H2A.Z and H4 acetylation mechanisms	− ↑ H2A.Z and H4 acetylation − ↑ SZ-associated transcriptional signature	[192]
	Wistar Han rats	− JQ1	− Sensorimotor gating − Recognition memory	− ↑ Prefrontal cortex development	[193]

Genes such as *Arc*, *Bdnf*, *Gria1,* and *Gria2*, whose expression was reduced by BET inhibitors, were elevated in reward-related brain regions by amphetamine and/or nicotine exposure [194,195,196]. Although JQ1 does not alter morphine and oxycodone CPP, another study showed that histone acetylation modifications, which are recognized by BET reader proteins, are elevated in the post-mortem brains of chronic opioid abusers, and injections of JQ1 in the dorsal striatum were capable of reducing heroin self-administration in rats [182]. In animal models of SUD, experimental evidence was generally collected by using pan-BET inhibitors such as JQ1, which bind to both BET BrDs (BrD1 and BrD2). Because pan-BET inhibitors may be associated with relevant adverse events, a clinically tested BrD2-specific BET inhibitor, RVX-208, was used in a subsequent study by Sartor’s research group. Similar to JQ1, this inhibitor dose-dependently reduced cocaine CPP in male and female mice. In other behavioral experiments, RVX-208 did not affect distance traveled, anxiety-like behavior, or NOR memory. Moreover, treatment with RVX-208 decreased the transcriptional levels of several cocaine-induced genes in the NAc in a sex-dependent manner. RVX-208 showed effects on gene expression in stimulated primary neurons compared to JQ1 but did not elicit a distinct transcriptional response in non-stimulated neurons. These researchers suggested, therefore, that targeting specific BET domains may represent a safer therapeutic approach to attenuate neurobehavioral adaptations mediated by cocaine [197].

Even though no studies analyzed the functional involvement of BET proteins in alcohol-induced behaviors, transcriptional levels of BRD3 and BRD4 were found to be decreased in the dorsal medial prefrontal cortex of alcohol-addicted rats [198]. These results highlight BET proteins as novel regulators of drug-induced neuroadaptation and suggest that BET modulation can be exploited in the development of therapeutic strategies for the treatment of SUD-related behaviors [65].

Several works displayed a possible connection between BET activity and other psychiatric disorders. Single-nucleotide polymorphisms associated with schizophrenia, for example, link BRD4 to an increased susceptibility to this disorder [199]. 

In a recent work, Farrelly and colleagues investigated histone acetylation in schizophrenia, identifying BRD4 as a possible target for treatment. In this study, the authors used fibroblasts derived from schizophrenia patients and healthy subjects, reprogramming them into human induced pluripotent stem cells, and then differentiated into neurons in vitro. By a proteomics approach, increased acetylation of the histone variant H2A.Z and of H4 were observed in neurons of patients with schizophrenia. These data were also confirmed in the postmortem human brain. Biochemical analysis, including X-ray crystallography, revealed that BRD4 is a H2A.Z acetylation reader. Notably, BRD4 pharmacological inhibition hampered the interaction between H2A.Z acetylation and BRD4, thus improving the transcriptional signature associated with schizophrenia in patient-derived neurons [192]. 

Another report evaluated the involvement of BET proteins in a model of schizophrenia induced by prenatal administration of methylazoximethanol (MAM). In particular, the authors demonstrated that the alterations associated with schizophrenia induced by MAM treatment were found only in males; on the contrary, JQ1 administration in the adolescent period affected behavioral responses and altered molecular and proteomic scenarios in the prefrontal cortex of both sexes. These results led to the hypothesis that JQ1 treatment during adolescence could affect the prefrontal cortex development [193].

The effects of BRD4 inhibition have also been investigated on depression/anxiety-like behaviors and spatial and fear memory in a model of PTSD using contextual and cue fear tests, the sucrose preference test, open-field test, elevated plus maze test, and Y-maze test in mice [200]. Inescapable foot shocks (IFSs) with a sound reminder in 6 days induced BRD4 expression in the prefrontal cortex, hippocampus, and amygdala, causing alterations of IEG. Furthermore, IFS induced depression, anxiety-like behaviors, and impairments in memory and spatial learning. JQ1 treatment counteracted freezing time in contextual and cue fear tests, nullified the behavioral impairments, and recovered IEG expression levels. Taken together, these reports indicate that BET inhibitors exhibit encouraging effects in several in vivo models of psychiatric disorders.

## 5. Conclusions

Regulation in gene expression is based on a wide interplay of different signal transduction pathways, leading to positive or negative transcriptional control. Many of these pathways modulate the activation and binding of TFs to DNA response elements. On the other hand, other pathways act on chromatin accessibility through epigenetic modification on histones exerted by writers and erasers (such as HAT and HDAC) [18]. Among the histone code readers, BET proteins play crucial roles in the regulation of gene expression, as they recognize Kac and favor the recruitment of the transcriptional machinery, thus regulating several cell processes in different tissues [25], including the brain. Recent evidence demonstrated that BET family members guarantee the expression of genes encoding neurotrophic factors, neurotransmitter receptors, ion channels, and many other proteins involved in synaptic plasticity [15,16,18,23]. Accordingly, different studies showed that BET proteins are expressed in all brain cells and their relative abundance depends on the cellular subtype [65], highlighting that variable expression patterns could be associated with specific cell functions and response to external *stimuli*. Despite this information, the expression of BET proteins in diverse brain regions is still elusive, and their role was exclusively dissected in selected neuronal subtypes [66]; further molecular and functional characterization is needed to obtain deeper insight into the role played by BET proteins in brain homeostasis. 

Since BET proteins are emerging as crucial epigenetic players in CNS physiology, it is not surprising that alterations in their activity may be associated with several neuropathological conditions. From this perspective, the use of BET inhibitors in preclinical research facilitated the dissection of physiopathological mechanisms in cell culture and animal disease models, providing the rationale to target BET proteins as a novel therapeutic strategy. Concerning neurodegenerative conditions, only few data are available about the prospective impact of BET inhibition in HD and PD, whereas most research has focused on AD. Even though few in vitro studies pointed out a detrimental role of BET inhibition in AD pathology [146], a growing body of evidence underlines that BET blockade could counteract cognitive dysfunction and neuroinflammation in different cellular and animal models [23,143,145]. The discrepancy among these studies may be explained considering the specific experimental models, as well as drug dosage and treatment protocols. BET proteins also affect brain development; it has been observed that BET protein activity is essential to assure proper neurodevelopment [19]. Conversely, BET inhibition significantly ameliorates neuronal defects in experimental models of neurodevelopmental diseases, such as RTT and FXS. Notably, these disorders are characterized by abnormal epigenetic landscapes, which lead to hyperactive transcription [102,103]. Thus, BET modulation is not beneficial or detrimental for neurodevelopment per se, as its impact is strictly dependent on the physiopathological context. It is interesting to emphasize the dose-dependent effects of BET inhibitors tested in the aforementioned conditions. For instance, in the mouse model of FXS, low-dose treatment with JQ1 alleviated disease-associated phenotypes [102]; however, a high dose of JQ1 could impair the processes ensuring memory formation [15]. Similarly, only low doses of JQ1 were found to improve the RTT phenotype in vivo [103]. Further studies are needed to further investigate the therapeutic opportunities based on BET targeting in the pathophysiological processes that influence neurodevelopment.

Furthermore, BET proteins have also been associated with cognitive dysfunction and neuropsychiatric diseases, such as SUD, FTD, and schizophrenia [16,153,192].

Collectively, these findings help to shed light on BET proteins’ activity and their possible inhibition when approaching disease treatment. However, several issues need to be addressed. As already reported, more efforts should be made in terms of fundamental research to deeply dissect the role of each BET protein in brain cells; a better comprehension of the molecular, cellular, and functional processes is indeed essential to properly comprehend the contribution of BET proteins in CNS physiopathology and the possible employment of BET inhibition as effective therapeutic strategy. In this context, it will also be important to assess the pharmacological efficacy of the selective manipulation of a specific BrD. Indeed, BrD1- or BrD2-selective BET inhibitors have been recently developed and showed encouraging therapeutic effects with less adverse events [201]. 

The addition of missing pieces in the intricate puzzle depicting the involvement of BET proteins in the brain will certainly be useful to evaluate promising therapeutic opportunities for CNS disorders.

## Figures and Tables

**Figure 1 biomedicines-11-00750-f001:**
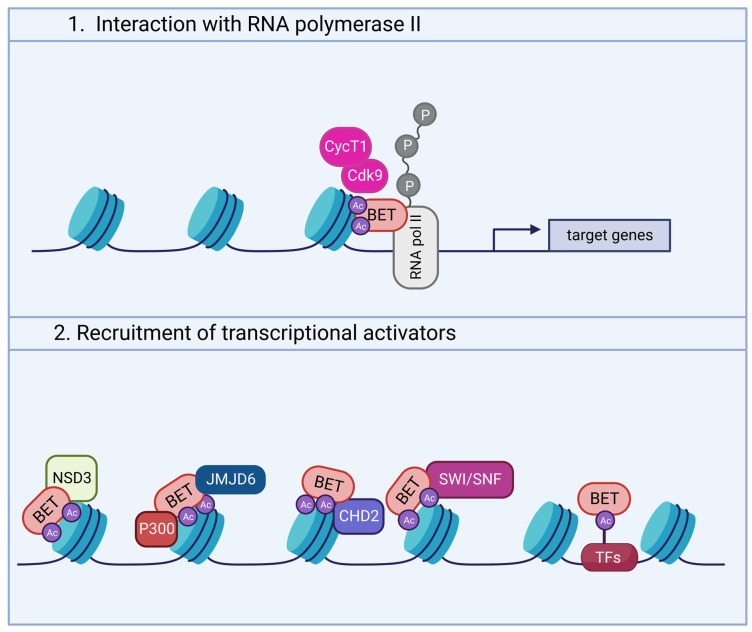
Transcriptional regulatory mechanisms mediated by BET proteins. (**1**) Binding of BET proteins to acetylated histones results in interaction with the p-TEFb elongation complex (CDK9/CycT1), promoting phosphorylation of the C-terminal domain of RNA polymerase II and transcription of target genes. (**2**) BET protein-mediated recruitment of transcriptional activators, such as histone demethylase JMJD6, CBP/p300 HAT, methyltransferase NSD3, and the nucleosome remodeling complexes SWI/SNF and CHD2, leads to changes in chromatin structure. Furthermore, BET proteins are also able to bind other acetylated proteins such as TFs, modulating transcription. This figure is created with BioRender.

**Figure 2 biomedicines-11-00750-f002:**
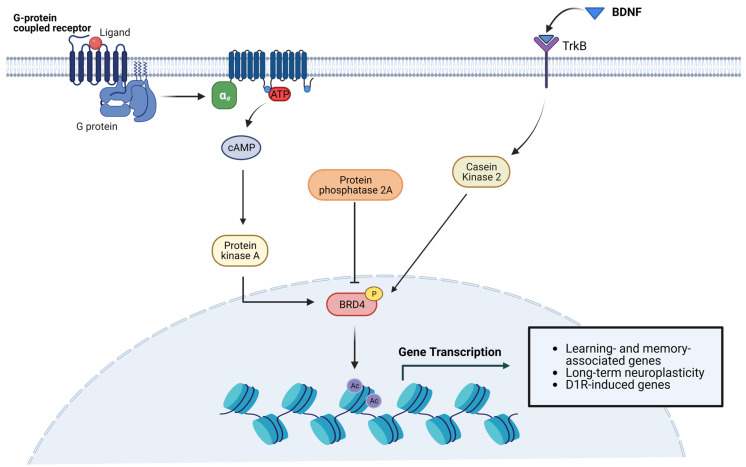
Proposed model for BRD4 activation in neurons. Gs protein-coupled receptors and TrkB activation recruit cAMP/PKA axis and casein kinase 2 (CK2), respectively, promoting BRD4 phosphorylation and subsequent Kac recognition on chromatin. This event results in the transcriptional activation of genes involved in learning and memory, neuroplasticity, as well as upregulation of genes mediated by D1R. Protein phosphatase 2A inhibits BRD4 recruitment to chromatin through dephosphorylation [15,66,67,68]. This figure is created with BioRender.

**Figure 3 biomedicines-11-00750-f003:**
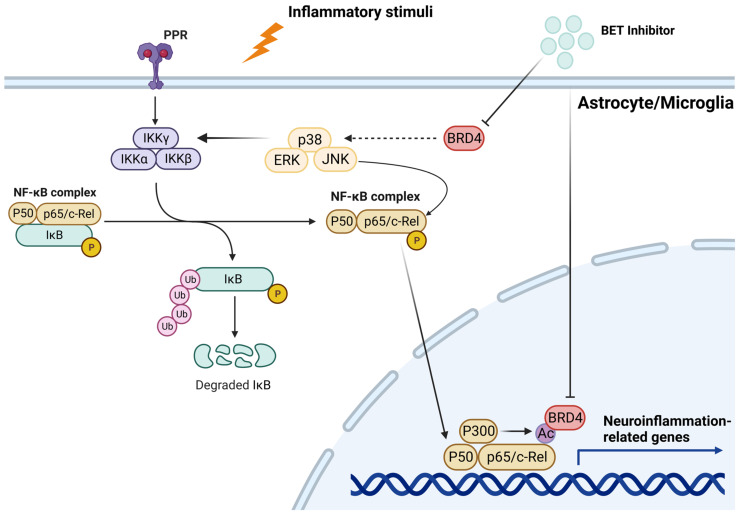
BRD4 interacts with NF-kB to regulate the expression of inflammation-associated genes. Activation of pattern recognition receptors (PPR) by inflammatory stimuli determines IKK α/β/γ complex recruitment resulting in IkB phosphorylation and later degradation through ubiquitin/proteasome-dependent mechanisms. IkB downregulation leads to NF-kB subunit p65/c-Rel phosphorylation by MAPK members (p38, ERK and JNK), promoting its nuclear localization. Nuclear NF-kB binds to target gene promoters recruiting acetyltransferase CBP/p300 complex. p65/c-Rel subunit is later acetylated at Lys310, leading to BRD4 recognition of acetylated NF-kB and upregulation of genes involved in neuroinflammation. BET inhibitors exert anti-inflammatory effects by preventing BRD4 binding to acetylated NF-kB and inhibition of MAPK activation through indirect BRD4-dependent mechanisms in both astrocytes and microglia [123,124]. This figure is created with BioRender.

## Data Availability

Not applicable.

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
