# Peer review of "Bromodomain and Extra-Terminal Proteins in Brain Physiology and Pathology: BET-ing on Epigenetic Regulation"

_biomedicines, 2023, doi:10.3390/biomedicines11030750_

Round 1

Reviewer 1 Report

Comments to the Author:

This review article, focuses on recent evidence on the involvement of BET proteins in brain physiology and pathology, highlighting their pharmacological potential. This topic is very interesting, and this manuscript greatly highlights the role of BET proteins and their potential impact on physiology and disease.

Overall, I just feel that talking about brain pathology and disease without discussing normal ageing is a caveat. Age-driven cognitive impairment and neurodegeneration should be addressed here, even if in a simple manner. I only have some questions that hopefully might help the authors improving the impact of their manuscript.

1) BRD4 was found to be related to dopamine-induced signaling. Considering the specific loss of dopaminergic neurons in PD, does this BET protein play a distinct role in the pathophysiology of this disease? Could you further discuss this topic as in your discussion in the neurodegenerative section, this feels a bit incomplete.

2) Considering the role of BET proteins in neuronal differentiation and neurodevelopment, are there any reports concerning these proteins and adult neurogenesis? Since the loss of the neurogenic potential is one hallmark of neurological diseases, more specifically, psychiatric, and neurodegenerative, this could be a topic worth mentioning.

3) In the manuscript the authors refer to clinical studies (in humans) using BET inhibitors. However, the manuscript might benefit from the creation of a separate section discussing BET targeting (inhibition and upregulation) in human disease, as this will highlight the translation potential of these molecules. Furthermore, what side effects were observed in these clinical studies, besides a mild cognitive improvement?

4) Following the previous point, you could also discuss whether mutations or polymorphisms in human genes encoding BET proteins are involved in disease.

5) Considering ageing as a major trigger for the development of neurological conditions and neurodegenerative diseases, what happens to BET protein levels and gene expression during the ageing process? Consider adding a section referring to ageing and age-related neurological conditions, highlighting the potential impact of BET proteins.

6) Relative to the regulation of neuroinflammation, there is a fine line in the balance of a pro- versus anti-inflammatory state and health versus disease. Could you further discuss the impact of targeting inflammation in disease? Is it a linear approach?

7) Considering that the application of pharmacological modulation of BETs, you should insert a section “applications and limitations”. Here you should discuss the potential and real applicability of these approaches, which are mentioned throughout the document. Furthermore, the limitations should also be discussed, is it safe to target a protein involved directly in transcription regulation? What member of the BET family might be more promising to target for neurological diseases?

8) In section 3.1, the authors state “Li and colleagues demonstrated that pharmacological inhibition of BET proteins by JQ1 resulted in the activation of a specific transcription program that promotes neuronal differentiation of neural progenitor cells (NPCs) isolated from mouse cortices, inhibiting the differentiation towards a glial phenotype.”. Although these is explained in the next statements, this sentence is confusing. Consider rephrasing.

9) The abstract should be improved as it is not aligned with the quality of the work.

Author Response

Reviewer 1

- “This review article, focuses on recent evidence on the involvement of BET proteins in brain physiology and pathology, highlighting their pharmacological potential. This topic is very interesting, and this manuscript greatly highlights the role of BET proteins and their potential impact on physiology and disease.”

Reply by the authors: We sincerely thank the reviewer for the positive remarks on our manuscript.

- “1) BRD4 was found to be related to dopamine-induced signaling. Considering the specific loss of dopaminergic neurons in PD, does this BET protein play a distinct role in the pathophysiology of this disease? Could you further discuss this topic as in your discussion in the neurodegenerative section, this feels a bit incomplete.”

Reply by the authors: We really appreciate the reviewer’s observation. As correctly stated, BRD4 has been implicated in dopaminergic functioning, thus leading to hypothesize that BET modulation could be involved in PD. To our knowledge, there is no literature data linking BET proteins and PD. We found only two papers linking BET protein activity to the onset of levodopa-induced dyskinesia, which are already included in our review. In contrast, there are no data on the involvement of BET proteins in the pathogenesis of PD.

- “2) Considering the role of BET proteins in neuronal differentiation and neurodevelopment, are there any reports concerning these proteins and adult neurogenesis? Since the loss of the neurogenic potential is one hallmark of neurological diseases, more specifically, psychiatric, and neurodegenerative, this could be a topic worth mentioning.”

Reply by the authors: We sincerely appreciate this reviewer's comment. The involvement of BET proteins in neuronal differentiation and neurodevelopment leads to the hypothesis that this class of epigenetic regulators may influence adult neurogenesis. However, to date, there is no published research specifically addressing this issue.

- “3) In the manuscript the authors refer to clinical studies (in humans) using BET inhibitors. However, the manuscript might benefit from the creation of a separate section discussing BET targeting (inhibition and upregulation) in human disease, as this will highlight the translation potential of these molecules. Furthermore, what side effects were observed in these clinical studies, besides a mild cognitive improvement?”

Reply by the authors: In order to be as comprehensive as possible, we included both preclinical and clinical observations in our review. To date, we have found that there are only few clinical studies on the use of BET inhibition in brain physiopathology. We have included these reports in our review, but the information is still too limited to warrant a separate section. Regarding side effects, it should be mentioned that the effect of BET inhibitors is strictly dependent on the type of inhibitor and the dose schedule employed in the study, as well as on the epigenetic landscape, which may vary greatly between different pathological conditions. As a result, the pattern of potential adverse events could be very different, as well as their severity. Given the limited information and the large number of variables influencing the onset of adverse events induced by BET inhibitors, we prefer not to include this aspect as it may be confusing for the reader.

- “4) Following the previous point, you could also discuss whether mutations or polymorphisms in human genes encoding BET proteins are involved in disease.”

Reply by the authors: To our knowledge, mutations/polymorphisms in genes encoding for BET proteins have been discovered only in the context of cancer biology (Lori et al., 2016, PLoS One. doi: 10.1371/journal.pone.0159180). As there is no substantial information available on the potential occurrence of mutations/polymorphisms of BET-encoding genes in the context of brain physiopathology, we prefer not to mention this aspect as it falls outside the scope of the review article.

- “5) Considering ageing as a major trigger for the development of neurological conditions and neurodegenerative diseases, what happens to BET protein levels and gene expression during the ageing process? Consider adding a section referring to ageing and age-related neurological conditions, highlighting the potential impact of BET proteins.”

Reply by the authors: Thanks to the reviewer for this very interesting point of reflection. Unfortunately, to our knowledge, there is no information on the modulation and/or role of BET proteins in the aging brain.

- “6) Relative to the regulation of neuroinflammation, there is a fine line in the balance of a pro- versus anti-inflammatory state and health versus disease. Could you further discuss the impact of targeting inflammation in disease? Is it a linear approach?”

Reply by the authors: The regulation of neuroinflammation and the balance between pro- and anti-inflammatory conditions are of particular interest. However, we believe that this topic is too complex to be discussed in detail and deserves a dedicated research article. Moreover, neuroinflammation is only one of the aspects we have discussed in our review article.

- “7) Considering that the application of pharmacological modulation of BETs, you should insert a section “applications and limitations”. Here you should discuss the potential and real applicability of these approaches, which are mentioned throughout the document. Furthermore, the limitations should also be discussed, is it safe to target a protein involved directly in transcription regulation? What member of the BET family might be more promising to target for neurological diseases?”

Reply by the authors: We thank the reviewer for this interesting suggestion. The use of a drug always depends on the balance of risks and benefits. This is particularly true for epigenetic drugs, which can of course have serious side effects. On the other hand, their use could be useful to fight life-threatening pathologies. As an instance, the HDAC inhibitor panobinostat is currently used in clinical practice to treat multiple myeloma. Another advantage of epigenetic drugs, including BET inhibitors, is that they may have the potential to address multiple aspects of complex diseases by affecting the transcription of different target genes. Unfortunately, there is limited information available on this aspect in the context of brain disorders, and we therefore prefer to be more cautious in our considerations due to the lack of robust published data. However, we have already highlighted the importance of increasing knowledge in order to better identify the effective use of BET inhibitors in the treatment of human brain disorders (for details, see the "Conclusion" section).

- “8) In section 3.1, the authors state “Li and colleagues demonstrated that pharmacological inhibition of BET proteins by JQ1 resulted in the activation of a specific transcription program that promotes neuronal differentiation of neural progenitor cells (NPCs) isolated from mouse cortices, inhibiting the differentiation towards a glial phenotype.”. Although these is explained in the next statements, this sentence is confusing. Consider rephrasing.”

Reply by the authors: According to the reviewer’s suggestion, we rephrased the sentence as it follows: “Li and colleagues demonstrated that inhibition of BET proteins by JQ1 promotes differentiation of mouse cortical neural progenitor cells (NPCs) towards a neuronal, rather than a glial, phenotype”

- “9) The abstract should be improved as it is not aligned with the quality of the work.”

Reply by the authors: According to the reviewer’s suggestion, we revised the “abstract” section by including additional information.

Reviewer 2 Report

Dear Noemi Martella and colleagues,

your review on the role of bromodomain containing BET proteins in neuronal development and disorders is very comprehensive and easy to follow.

I have only a few comments:

page 3: it is not clear which protein(s) deliver(s) the kinase activity in the context of BRD4

Since no reader keeps all the biological functions of the different proteins/genes in his or her mind, it is helpful to mention their function either in the text (e.g. demetylase JMJD6 or in brackets after the protein/gene name (e.g. TuJ1 ( beta-tubulin III)). Sometimes this information is given in the text, some times it has been omitted. Please add this information for all Protein/Genes.

I find that the section on the inhibitors on page 4 needs more background (i.e. mode of action) as they play an important role later in the text.

In Tables 1 & 2, please add the full names for the disease abrreviations in a table legend

In Figure 1 top panel 1; I would give BET a more prominent colour as the pale yellow is difficult to see - ideally the same colour in all figures (ie light red)

Conclusion: I am not sure whether BET proteins are "key regulators". While they are certainly important to read the correct acetylation code, they are only part of protein complexes assembling at the chromatin. The key regulator would the molecule that initiates complex assembly.

Author Response

Reviewer 2

- “Dear Noemi Martella and colleagues,

your review on the role of bromodomain containing BET proteins in neuronal development and disorders is very comprehensive and easy to follow.

I have only a few comments:”

Reply by the authors: We are very grateful to the reviewer for the appreciation of our manuscript.

- “page 3: it is not clear which protein(s) deliver(s) the kinase activity in the context of BRD4”

Reply by the authors: We thank the reviewer for this observation. Interestingly, BRD4 acts as an atypical kinase. This detail is now included in the main text (Please see paragraph 2: “Subsequently BRD4, in association with P-TEFb, displaces the 7SK/HEXIM complex and, acting as an atypical kinase, phosphorylates the Ser2 in the C-terminal region of RNApol-II [40,46,47]”)

- “Since no reader keeps all the biological functions of the different proteins/genes in his or her mind, it is helpful to mention their function either in the text (e.g. demetylase JMJD6 or in brackets after the protein/gene name (e.g. TuJ1 ( beta-tubulin III)). Sometimes this information is given in the text, some times it has been omitted. Please add this information for all Protein/Genes.”

Reply by the authors: We thank the reviewer for this suggestion. When necessary, this information is now added in the text.

- “I find that the section on the inhibitors on page 4 needs more background (i.e. mode of action) as they play an important role later in the text.”

Reply by the authors: We are really grateful to this reviewer’s comment, which allow us to significantly improve paragraph 2. According to the suggestion, we added additional information on BET inhibitors (please see paragraph 2: “The first pan-BET inhibitors discovered were 3-methyltriazolothienodiazepines and methyltriazolobenzodiazepines. Structural studies have shown that members of this class of small molecules bind to the acetyl-lysine binding pocket of BET BrDs. In particular, the triazolo ring functions as an acetyl-lysine mimetic moiety, mimicking the hydrogen bond determined by the acetyl-lysine carbonyl to a conserved asparagine and a water-mediated hydrogen bond to a conserved tyrosine. A specific feature of BET BrDs is the WPF shelf, which is often targeted by aromatic or hydrophobic moieties of BETi, thereby conferring great poten-cy and selectivity to BET domains. JQ1 is the prototype pan-BET inhibitor. Importantly, this compound is still being used successfully for proof-of-concept studies in preclinical research. Following the discovery of JQ1, frag-ments containing a dimethyl isoxazole core led to the synthesis of new and different classes of pan-BET inhibitors such as iBET151, the sub-nanomolar inhibitor HJB97 and other molecules that have successfully entered clinical trials. To date, more than 600 crystal structures of BET BrDs have been registered in the Protein Data Bank (PDB, https://www.rcsb.org/). The different roles of BET proteins in maintaining cellular homeostasis led to speculation that BETi, to be successful in clinical practice, should be selective for a specific BrD. Interestingly, recent studies have shown that selective targeting of BD1 or BD2 is possible and could be an effective strategy to achieve therapeutic potential by limiting off-target effects. For example, RVX-208 has been shown to interact with sol-vent-exposed residues that are conserved in BD2 but different in BD1. These variations in specific amino acid residues were essential for the development of novel BD2-selective inhibitors such as ABBV-744. On the other hand, the BD1 selectivity of BETi was achieved by optimizing the interaction with the unique aspartate/lysine res-idues located in the BC loop. For more information on BETi, please see the review by Schwalm and Knapp (2022).

- “In Tables 1 & 2, please add the full names for the disease abrreviations in a table legend”

Reply by the authors: We thank the reviewer for this suggestion. Disease abbreviations are now integrated in the table legends.

- “In Figure 1 top panel 1; I would give BET a more prominent colour as the pale yellow is difficult to see - ideally the same colour in all figures (ie light red)”

Reply by the authors: We thank the reviewer for this suggestion. We standardized the color attributed to the BET proteins in all figures, choosing a light red as recommended.

- “Conclusion: I am not sure whether BET proteins are "key regulators". While they are certainly important to read the correct acetylation code, they are only part of protein complexes assembling at the chromatin. The key regulator would the molecule that initiates complex assembly.”

Reply by the authors: We understood the reviewer’s point. Thus, “key regulators” is now changed to “crucial epigenetic players”.

Reviewer 3 Report

The manuscript "Bromodomain and Extra-Terminal Proteins in Brain Physiology and Pathology: BET-ing on Epigenetic Regulation" written by Martella N, Pensabene D, Varone Michela, Colardo M, Petraroia M, Sergio W, La Rosa P, Moreno S and Segatto M. presents the structure and function of BET proteins, bromodomain containing proteins which have a role in recruitment of transcriptional activators and chromatin remodelers to gene promoters. The manuscript is a review which, after short introduction into the topic and description of the BET structure and function, presents the role of BET proteins in normal brain physiology and then in neuropathological conditions (from neurodevelopmental disorders, neuroinflammation to neurodegenerative and neuropsychiatric disorders).

The manuscript is comprehensive, well written and organized. Great part of the content is an overview of different experiments on mice and rats used as models for different diseases and treated with BET inhibitors. The accent is put on animal behavior in specific tests, so on brain function. Therefore, other diseases (such as tumors) were not so much in the focus.

Minor comments:

page 10: cohesinopathies

page 12: coding

page 18: ALS

Tables need explanation of the abbreviations

Author Response

Reviewer 3

- “The manuscript "Bromodomain and Extra-Terminal Proteins in Brain Physiology and Pathology: BET-ing on Epigenetic Regulation" written by Martella N, Pensabene D, Varone Michela, Colardo M, Petraroia M, Sergio W, La Rosa P, Moreno S and Segatto M. presents the structure and function of BET proteins, bromodomain containing proteins which have a role in recruitment of transcriptional activators and chromatin remodelers to gene promoters. The manuscript is a review which, after short introduction into the topic and description of the BET structure and function, presents the role of BET proteins in normal brain physiology and then in neuropathological conditions (from neurodevelopmental disorders, neuroinflammation to neurodegenerative and neuropsychiatric disorders).

The manuscript is comprehensive, well written and organized. Great part of the content is an overview of different experiments on mice and rats used as models for different diseases and treated with BET inhibitors. The accent is put on animal behavior in specific tests, so on brain function. Therefore, other diseases (such as tumors) were not so much in the focus.”

Reply by the authors: We are very grateful to the reviewer for the appreciation of our manuscript.

- “Minor comments:

page 10: cohesinopathies

page 12: coding

page 18: ALS

Tables need explanation of the abbreviations

Reply by the authors: We thank the reviewer for the suggestions. We carefully addressed all the minor concerns.
